# Lesson Study as a Means to Change Secondary Preservice Physics Teachers' Practice in the Use of Multiple Representations in Teaching

Teresa Conceição *, Mónica Baptista and João Pedro Ponte

Institute of Education, University of Lisbon, 1649-013 Lisbon, Portugal; mbaptista@ie.ulisboa.pt (M.B.); jpponte@ie.ulisboa.pt (J.P.P.)
* Correspondence: mariaconceicao@campus.ul.pt

**Abstract:** Multiple representations, such as experimental data, schemas, tables, and graphs, are an essential resource in science teaching. However, their use in the classroom typically poses a challenge for preservice teachers. The aim of this research is to examine changes in the practices of a group of preservice teachers regarding the use of multiple representations in the teaching of kinetic energy to 9th grade students, when this training is included in their initial teacher education program. For this purpose, a collaborative, reflexive, and student-learning centered approach, namely, a lesson study with three cycles, was implemented. A descriptive and content analysis for qualitative data collected showed improvement in the practices of the preservice teachers, namely on the representations both of the event that represents the "real" world, as well as of the scientific concepts. The results obtained contribute to deepening the knowledge on the use of multiple representations by preservice teachers, as well as to increasing the knowledge on using lesson study to develop the ability to use multiple representations during initial teacher education.

**Keywords:** initial teacher education; lesson study; multiple representations; preservice teachers; professional development; science teacher education

## 1. Introduction

Multiple representations (MR) consist in using two or more representations (such as diagrams, illustrations, schemas, tables, graphs, algebraic equations, web-based simulations, artifacts, and other non-textual forms) in teaching a curricular topic [1,2]. MR are also present in other situations, scientific and nonscientific, and play a fundamental role in understanding scientific concepts and technological applications in daily life [3] as well as in scientific research [4].

MR have been under intense scrutiny in the field of education since the end of the 20th century [2,5–8]. Several studies have shown that the use of MR in science teaching enables students to better conceptualize scientific concepts as they can make sense of real-life situations and common phenomena [3,5,9]. In addition, MR promote students' interest in learning scientific concepts by providing opportunities for linking abstract concepts with concrete phenomena [10,11]. Moreover, using MR helps students and teachers to improve the understanding of the concept or the phenomenon because each representation focuses on a different aspect of the concept or the phenomenon. Therefore, they are an excellent resource for students to learn science concepts [3,5].

Educational reform documents in the United States of America [12] and Europe [13] state that it is important to create learning opportunities from initial teacher education to build teacher skills related to the use of MR. These skills include knowing the best representations for teaching and how to use them to support the learning of all students. Despite the strengths of MR, empirical studies have shown that teachers [5,14] and preservice teachers

(PSTs), even those with undergraduate degrees in science [8,15], have difficulty using them in science teaching.

Several studies showed that long-term inquiry-based teacher education contexts are favorable during the initial teacher education for PSTs to learn using representations [6,8,15–17]. One such teacher education approach is lesson study (LS), a reflective and student-centered teacher professional development process. This approach involves teachers constantly questioning about teaching practice in order to improve it. Previous studies in LS showed promising results as a learning process of PSTs [18–22], although requiring support by the teacher educators [23,24].

The knowledge on the LS applicability in developing PSTs' ability to use MR in physics teaching is still scarce. This paper aims to make a contribution in this regard. Thus, the aim of this research is to examine changes in the practices of a group of PSTs regarding the use of MR in the teaching of kinetic energy at 9th grade. For this purpose, a LS with three cycles was implemented based on the car crash test as the "real" world event. A car crash test is a real-life situation and a common phenomenon that helps students link abstract concepts with concrete phenomena and enables them to better conceptualize scientific concepts [9].

## 2. Background

### 2.1. Multiple Representations in Physics Teaching

MR are recognized to be crucial in science. In science teaching, MR have been acknowledged for their potential in visualizing phenomena, thus facilitating the understanding of scientific concepts [3,5]. Researchers [25] have shown that MR help students make connections across physics variables by deepening their understanding of concepts. The use of MR also contributes to the multiplication of meanings that emerge from different forms of representation [26], that might be used in science teaching by combination, i.e., using various representations separately or superimposed, i.e., laid over other representations [27]. Indeed, physical phenomena can be translated through consecutive representations, i.e., MR, which may include, in increasing order of complexity, images, tables, graphs, equations, and text [6]. As such, to support students' understanding of the world as seen by science, it is important to introduce to students the skills and processes that form the methodology and vocabulary of the scientist [28]. This can be achieved using MR of realistic information and helping students explain the scientific concepts that can make them wonder about real-world situations in an ongoing interaction between teacher and students and among students. These interactions include production, reading, transformation, and evaluation of MR [3]. In the scope of this study, MR means the products of a procedure (such as a table and a graph) and the tools for representing a phenomenon. Furthermore, constructing MR actively engages students in their learning and develops their thinking, predicting, making claims, understanding, and representing skills [29–31].

In fact, research showed the importance of including MR training in initial teacher education programs to support its use by PSTs in teaching science [32–36]. In one study [8], the participants were 15 preservice science teachers and they were enrolled in a course originally designed to allow PSTs to design and conduct open-ended experiments. The course emphasized the production of MR as a way of convincingly supporting claims when the PSTs presented their results. The results showed a significant increase in the number and type of MR made by the PSTs as the course unfolded. In addition, as the number of graphs, tables, and other sorts of complex representations made by the PSTs increased, the number of text-based representations decreased. In another study [6], 25 preservice secondary science teachers with a major either in science (the majority), mathematics (two), or art (one) were enrolled in a teacher preparation program with similar characteristics to the study in [8]. The study was designed to understand PSTs' scientific practices relative to the MR and transformations they were expected to teach according to the reform documents guidelines [37]. Researchers [37] showed that most PSTs had difficulties in relating a set of actual data in a Cartesian graph because of the scattering of the data. The authors mentioned that the relationship between the two variables was based on the comparison of

individual data pairs. Both studies showed the importance of including MR training in initial teacher education programs since PSTs significantly increased the number and type of representations (graphs, tables, and other sorts of complex representations) to which they resorted while the number of text-based representations decreased.

*2.2. Lesson Study*

LS is a teacher professional development process involving lesson planning and lesson enactment with observation and discussion, aiming to improve teaching and learning [38]. The characteristics of this approach promote teachers developing in-depth knowledge about the topic of the LS, as well as enhancing their teaching ability while students learn the topic [39].

LS implementation typically follows a model (schematically depicted at Figure 1), in which, usually, a small group of teachers of three to five [38,40] (i) consider long-term goals for students learning and development, as well as identify a curricular topic of interest; (ii) meet in work sessions to conduct a preparatory study to deepen their knowledge about the topic and to analyze the challenges of its teaching and the potential students' learning difficulties; (iii) collaboratively develop a very detailed lesson plan on the topic and a task to assist students' learning to be used in a research lesson; and (iv) one of the teachers teaches the research lesson while the others observe. The data collected during this lesson typically include observation and video recording. (v) Subsequently, during reflection, the participant teachers analyze in depth students' responses, paying attention to their learning progress and difficulties, reflecting on the teaching decisions made, and improving the teaching materials (i.e., the lesson plan and the students' task).

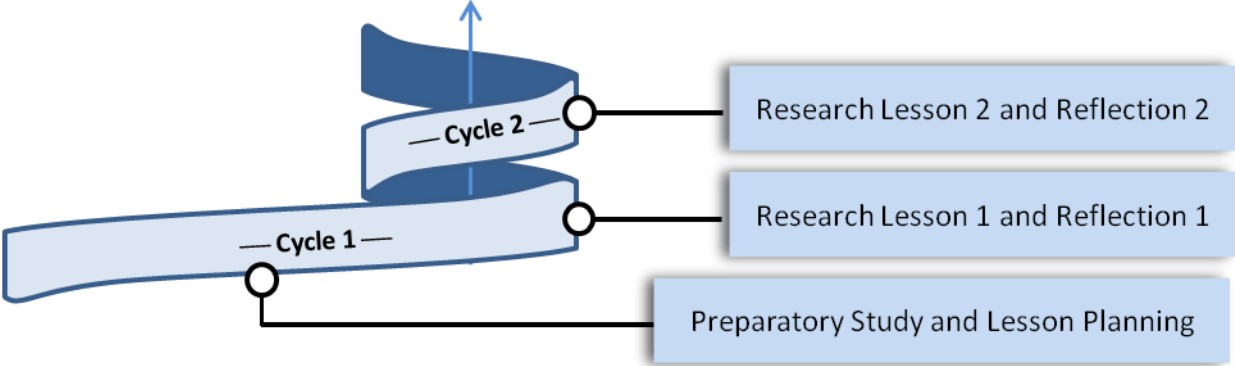

**Figure 1.** Typical model of the lesson study (LS).

Sometimes, a second cycle (or even more cycles) of the LS takes place (Figure 1). In the following cycle, that does not include the steps (i) to (iii) mentioned above, another teacher teaches the improved research lesson to another class of students and all participants reflect again on the teaching decisions made as well as refine the teaching materials in a second reflection.

However, in the literature, conducting sequential cycles in the LS is not consensual. For example, [38] argued that the objective of the LS is not to teach a "perfect" lesson but to educate teachers. Thus, that researcher does not consider it beneficial to implement more than one cycle. Apart from the controversy over the number of LS cycles, in the context of the initial teacher education, several investigations in LS have shown that it is advantageous for the professional knowledge of the PSTs to carry out more than one cycle [20,22,41–43].

Researchers [44] conducted a rare study regarding the use of MR in a microteaching LS. This investigation involved 65 future upper secondary physics and chemistry teachers who used a sensor, as well as multimedia simulation, to perform their MR. The research lessons were taught by the PSTs to their peers, who assumed the role of students. The results showed that the PSTs selected, essentially, multimedia simulations (i.e., PhET and Gizmos)

for their MR and that they found their involvement in a microteaching LS to be positive because allowed them to identify benefits and limitations of using web-based simulations in the teaching of scientific concepts. Despite the promising results of the study [44], the knowledge about the adoption of LS in the initial teacher education regarding the use of MR in the scientific teaching and particularly in physics teaching is still scarce. Indeed, from the results of a revision literature on Web of Sciences and Scopus with the words "lesson study", "representations", "graphs", "tables", "simulations", and "physics", only [44] appears.

## 3. Method

This section presents details of the participants in the investigation (Section 3.1), LS implementation (Section 3.2), and data collection and analysis (Section 3.3).

### 3.1. Characterization of the Participants in the Research

The participants in the research were the PSTs (n = 3)—PST1, PST2, and PST3—enrolled in a Master's degree in teaching science (physics and chemistry) at lower and upper secondary level (see more information in Table 1). The three PSTs, three cooperating teachers, one professor of the university (second author) that conducted the LS sessions, two educational researchers (first and third authors), and one physics researcher participated in the LS sessions (detailed in Section 3.2). The option to involve cooperating teachers [45,46] as well as researchers [8] stemmed from recognizing the relevance of these participants in the PSTs training.

**Table 1.** Characterization of the preservice teachers (PSTs).

| Preservice Teachers | Age | Gender | Education | Work Experience |
|---|---|---|---|---|
| PST1 | 42 | Male | Five-year first degree (major in chemistry with minor in physics) | Chemistry and physics teaching in a tutoring center |
| PST2 | 42 | Male | PhD in physics | Mathematics teaching in a private school |
| PST3 | 36 | Female | Five-year first degree (major in chemistry with minor in physics) | Chemistry teaching in a tutoring center |

### 3.2. Lesson Study Implementation

The sessions of the LS with the PSTs took place at the university and are schematically represented in Figure 2. The research lessons were taught by the PSTs in one of the classes of the cooperating teachers' schools. The topic selected for the LS was kinetic energy of the 9th grade curriculum and the "real" world event adopted was the car crash test. In a more general perspective, this topic encompasses the energy concept that is a crosscutting concept of the 12 years compulsory education.

Figure 2 schematizes the three cycles LS program implemented in this investigation. The program spanned approximately 51 h in total, each session taking three hours, except the three lesson reflections that took 90 min each. In concrete:

'Cycle 1' comprised 13 sessions, of which eight were devoted to the preparatory study of studying the MR as well as the topic to be taught with the MR (sessions 1 to 8); three to the lesson planning (sessions 9 to 11); one session consisted in the research lesson 1 (session 12), taught by PST1 (and observed by PST2 and PST3); and one to carry out the reflection 1 (session 13).

'Cycle 2' comprised one session that consisted in the research lesson 2 (session 14), taught by PST2 (and observed by the other two) on the same topic to a different class, as well as one session (session 15) to perform the respective reflection 2 that followed the same procedure of the previous Reflection 1 session.

'Cycle 3' comprised the two sessions that reproduced those of the 'cycle 2', i.e., research lesson 3 (session 16) and reflection 3 (session 17), except that the research lesson 3 (session 16) was taught by the third pre-service teacher, PST3.

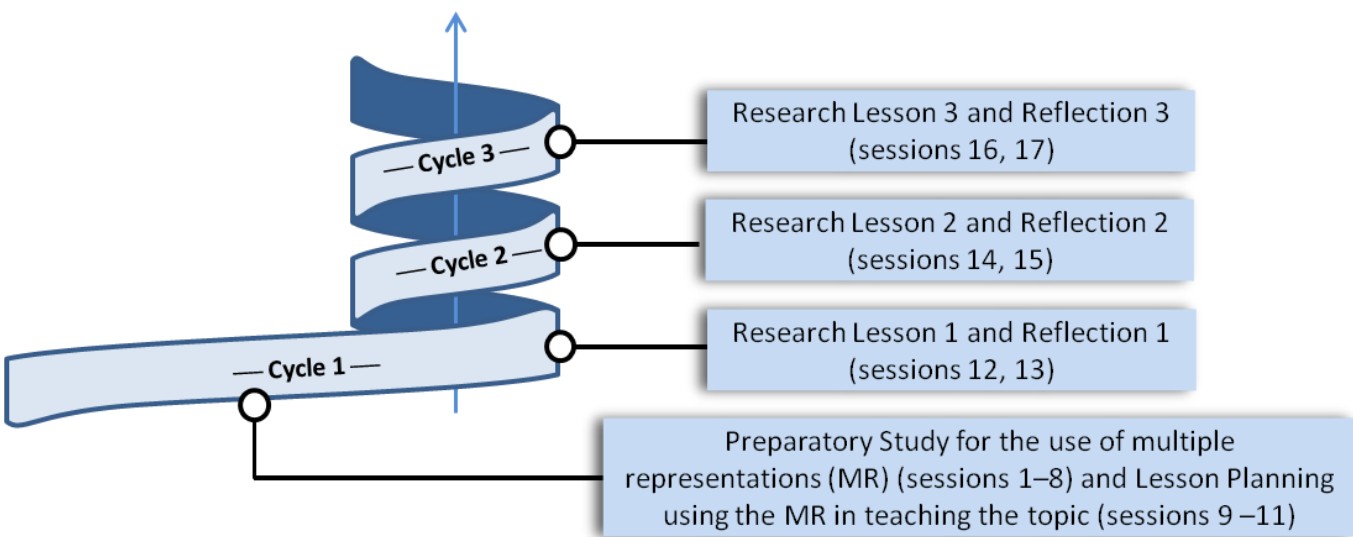

**Figure 2.** Scheme of the lesson study (LS) conducted.

As already mentioned, this investigation included eight sessions for the preparatory study (sessions 1–8) and three sessions (sessions 9–11) to perform lesson planning (Figure 2). This option helps PSTs to gain knowledge and insight into science as well as into student thinking [38,47]. The three cycles of the LS allowed each one of the PSTs teach a research lesson. Although the repetition of cycles is controversial in LS [38], in initial teacher education, it has been shown to be beneficial [41]. This investigation intended to follow the original Japanese LS [48] as much as possible, as it had to be adjusted to the PSTs' teacher education.

During the LS, to support the PSTs, teacher educators and researchers used inquiry strategies (such as PSTs asking questions related to students' difficulties and PSTs developing predications using students' evidence), promoting an active participation of PSTs in their own professional growth.

The professor from the university (second author) and the cooperating teachers participated in all sessions of the LS, supporting PSTs' learning. The physics researcher supported the PSTs with regard to the content of the topic and the use of MR in science and did not participate in all sessions of the LS. The researchers (first and third authors) collected data. In addition, one of the researchers (third author) adjusted the LS to the PSTs and supported the PSTs, mainly in the analysis of student responses and reasoning during the reflection sessions.

### 3.3. Data Collection and Analysis

The aim of this research is to examine changes in the practices of a group of PSTs regarding the use of MR in the teaching of kinetic energy at 9th grade. For this purpose, a LS with three cycles was implemented based on the car crash test as the "real" world event. In order to capture the PSTs' practices regarding the use of MR to teach kinetic energy, the preparatory study (sessions 1 to 8, Figure 2), the lesson planning (sessions 9 to 11), the research lessons (sessions 12, 14, 16), and reflections (sessions 13, 15, 17) were video recorded. In addition, the three educational researchers also participated in all sessions to take field notes. At the end of the LS, the PSTs were individually interviewed.

The video recordings of sessions 1 to 17 (about 51 h in total) and the interviews (about 5 h in total), were transcribed, and together with field notes of all sessions were analyzed to find changes that occurred in the PSTs' practices regarding the use of MR. For this purpose, a descriptive and content analysis [49] was performed for qualitative data. In particular, the transcripts of video records and field notes were read by the first author of this article. After that, the targeted text was segmented, representing ideas related with the research aim.

Each segment was assigned a code and clustered on a category, according to its features. Table 2 shows code examples for the category "Using the MR to support the teaching of the scientific concepts".

**Table 2.** Code example.

| Segmented Text (Data) | Code |
|---|---|
| With this graph, students visualize the relationship between kinetic energy and car's mass (video record). | Mass and graph |
| The algebraic equation relates the three variables, i.e., kinetic energy, mass, and speed (video record). | Variables and algebraic equation |
| The direction of acceleration of the mannequin's head is clearly visible in the photo sequence (video record). | Acceleration and photos |
| Students can calculate the force with which the mannequin's head hits the front seat from the acceleration value (video record). | Force and acceleration |
| The graph and photos complement each other (interview). | Photos and graph |

The categories were refined through a constant questioning and comparison of the data [50]. For example, the category "Using the MR to support the teaching of the scientific concepts" was redefined as follows: First, "Using the MR to visualize the kinetic energy", secondly, "Using the MR for the explanation of the kinetic energy", third, "Using the MR of the scientific concepts", and fourth, "Using the MR for the understanding of the scientific concepts". The final categorization was "Using the MR to support the teaching of the scientific concepts" as we have already mentioned. The codes and categories were driven from the data. Next, the second author analyzed the transcripts of the video records of sessions 9 to 17 and field notes of the corresponding sessions. Indeed, the PSTs only worked on the teaching practice about the kinetic energy in sessions 9 to 17. The option of the first author to analyzing the data from sessions 1 to 8 was due to the fact that the identification of changes in the practices of the PSTs with the MR (research objective) requires, in various situations, understanding what the PSTs already knew during the preparatory study (Session 1–8).

After that, based on the two descriptive categories that emerged from the data analysis, the second author analyzed the codes and created other codes if necessary. The two researchers compared their codes and discussed a consensus coding scheme, which was 89%; the proposal in [51] was considered for the calculation of reliability.

Taking consensual codification into account, the same two authors, independently, analyzed the interview transcripts considering the codes already defined, thus comparing their analysis. Disagreements and issues were discussed to reach a consensus. The consistency between the two authors was 91%. This procedure yielded the final categories of analysis. Table 3 presents the categories that emerged from the data analyses previously described in which the results (see Section 4) were organized as well as a description of the categories, Category 1 and Category 2.

**Table 3.** Categories that emerged from data analysis.

| Category | Description |
|---|---|
| 1. Changes in the MR that PSTs use to represent car crash tests as a "real" world event. | Evidence that PSTs changed the car crash tests representations, namely PSTs started using authentic data and other realistic information, instead non-realistic data and frictionless crash tests. |
| 2. Changes in the MR that PSTs use to support the teaching of the scientific concepts. | Evidence that PSTs changed the kinetic energy representations, namely PSTs started using a sequence of photos of the dummy head in different instants of the collision, a table elaborated by the students, and a Cartesian graph with experimental data, instead of algebraic equations and other complex representations, such as Cartesian graphs with fictional data. |

## 4. Results

This section presents some of the results obtained during the LS conducted (Section 3.2) as well as the rationale that supported the organization of results in the categories presented in Table 3.

*4.1. Changes in the MR That PSTs Used to Represent Car Crash Tests as the "Real" World Event (from Research Lesson 1 to Research Lesson 2)*

The following episodes were observed during research lesson 1 (session 12, Figure 3), in which PST1 explored, with the students, the kinetic energy phenomenon using a car crash-test event to illustrate.

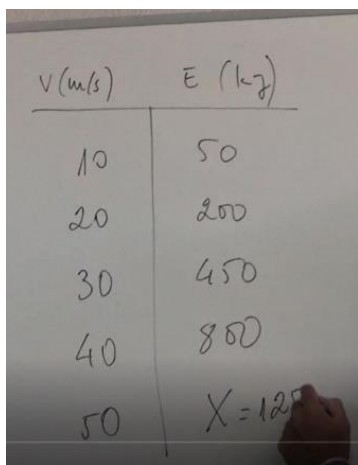

**Figure 3.** Made-up values of two variables of a car involved in a crash test event, speed (V, in m/s) and kinetic energy (E, in kJ) (Photo from research lesson 1).

In order to assist students in understanding the phenomenon, PST1 presented on the whiteboard a two-column table (Figure 3) with a made-up set of values of two variables, the car speed (V, expressed in m/s) and the corresponding kinetic energy (E, expressed in kJ) and asked the students to determine the unknown value of the kinetic energy of the car represented by the letter X in the table. After the students interacted in small-group conversations during approximately five min., PST1 promoted a whole-class discussion:

[00:04:30] PST1: You said that from here [points to number 50, in column 2, Figure 3] to here [changes the pointer to 200, below] there is an increase of 150. Then, how much is the increase from here to here [repeats the gesture from values 200 to 450 of the same column]?

[00:04:37] Student 16: 250.

[00:04:38] PST1: 250. And, from here to here? [moves the pointer from 450 to 800], 350. And, then . . . ? [points to 800 and moves the pointer to the unknown value represented with X in the line below] . . . [silence]. Thus, what is the value you obtain here? [points to the letter X and writes, X = 1250] (research lesson 1).

The transcript above shows that during the whole-class discussion, PST1 had difficulties in promoting the interpretation of the non-realistic event based on the set of values of the two variables (Vi, Ei), of a car involved in a crash-test presented in a two-column table (Figure 3). In fact, the encouragement for students to predict the unknown Ei value, X, from the sequence of the other Ei values presented in the table did not succeed well. Furthermore, PST1 used a non-realistic event since the (Vi, Ei) data presented were generated by an algebraic equation that reinforced for students the idea that the world is governed by mathematical models. This difficulty in relating the world and its representations was evident in many other episodes recorded in research lesson 1 (session 12, Figure 2) showing that in this research lesson 1, the PSTs were not able to represent a realistic event.

Later, during reflection 1 (Figure 2), the analysis of student responses resonated with PSTs making them to understand that "ideal" data did not help students understand the kinetic energy phenomenon applied to the crash test event, since students discussed data in the abstract, rather than discussing data applied to the specific event.

For example, during reflection 1, about students' reasoning when solving the task, PST2 and PST3 mentioned:

[01:10:32] PST3: The students used valid problem-solving strategies. One of the students divided the values of the kinetic energy by the corresponding values of the speed. They correctly determined the value of the kinetic energy, represented by X, but they were not able to describe how the kinetic energy of the car varied with its speed.

[01:10:57] PST2: One student that I observed did not even pay attention to the car speed column to determine the value of X. The relationship they [students] found did not include speed (reflection 1).

In fact, PST2 and PST3 recognized, in the transcript above, that the methods used by the students to try to determine the value of the kinetic energy X were correct but they were not able to relate the two variables, V and E, presented in the table (Figure 3) with a physical meaning. This difficulty of the students in assigning a physical meaning to the variables in a fictitious event was also registered in the interview of PST1:

[00:31:40]: In my class [research lesson 1], our data [non-realistic data] failed to represent the crash test event. They [students] only used the table to talk about numbers. When I said that the table represented an ideal situation, they did not understand. They know what they see and what they live with on a daily basis. This was one of the reasons why the students did not get involved in the discussion... they seemed to be feeling like they were in a math class! (Interview).

These analyses show that the students did not use the set of values (Vi, Ei) presented (Figure 3) to discuss the crash test event and to support a better understanding of this technological application of the kinetic energy phenomenon and that, rather, students only analyzed the values in the abstract. PST1 agreed with his colleagues' claims that "fictionalized" data did not represent the real-world situation well.

The transcript below exemplifies the PSTs' awareness of the importance of using realistic information to represent the event. This information was actually successfully used by PST2 when teaching research lesson 2, as PST2 predicted during reflection 1:

[01:29:08] PST3: They [students] do not understand the meaning of a frictionless situation because this does not represent their reality.

[01:29:16] PST2: The absence of friction is an abstract concept. If we want students to understand kinetic energy, we have to use real cases (reflection 1).

Thus, during reflection 1 (session 13), the PSTs searched for other ways of representing the kinetic energy phenomenon more realistically during research lesson 2. For this purpose, the PSTs considered that the best approach would be to use experimental data about crash-tests from a scientific article and use this information to represent the phenomenon of kinetic energy. Figure 4 presents an example of realistic crash tests included in the students' task of the research lesson 2 for two different models of baby chairs.

As such, PSTs showed that they changed the representations of the "real" world event they adopted to explore the kinetic energy phenomena to the students, since in research lesson 1, they used non-realistic data and in research lesson 2, they used experimental data and, therefore, realistic information. In fact, the PSTs observed that using data that represents a friction-free event (as PST1 did in research lesson 1) confused the students and the PSTs understood that they could overcome the situation presenting realistic real-world data (where objects perceptibly behave differently, i.e., they slow down). Accordingly, the PSTs improved the lesson plan for research lesson 2 as well as the students' task using a "real" crash test with experimental data.

In research lesson 3, the PSTs kept the same strategy of using a realistic crash-test because the solution found by the PSTs in reflection 1 and tested in research lesson 2 was shown to support student learning.

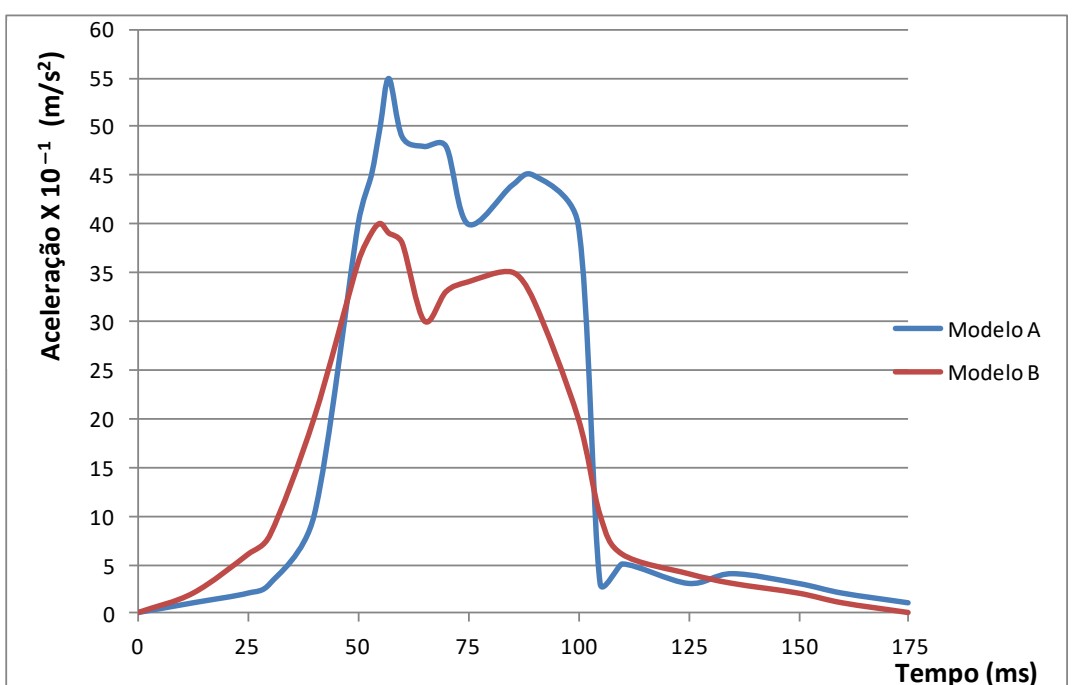

**Figure 4.** Realistic crash tests included in the students' task of research lesson 2 for two different models of baby chairs (in English, "Aceleração" is "Acceleration", "Tempo" is "Time", and "Modelo" is "Model").

*4.2. Changes in the MR That PSTs Used to Support the Teaching of the Scientific Concepts (from Research Lesson 1 to Research Lesson 2 and from Research Lesson 2 to Research Lesson 3)*

The PSTs changed the MR of the scientific concepts from research lesson 1 to research lesson 2 and from research lesson 2 to research lesson 3. Below, we present episodes that occurred during research lesson 1 and reflection 1 that made teachers aware of the need to change the representation of the scientific concepts to present in research lesson 2.

At a certain point of research lesson 1, while students were doing their task, PST1 asked them to select from a list of hypotheses (Figure 5), the algebraic equation that correctly related the kinetic energy (E) of the car with its mass (m) and speed (V) (the correct answer was option ii).

Considering the following list of algebraic equations, select the one from which you can determine the kinetic energy (E) of the car from its mass (m) and speed (V):

(i) $E = 0.5 \times m \times V$

(ii) $E = 0.5 \times m \times V^2$

(iii) $E = \frac{0.5}{m} \times V^2$

(iv) $E = \frac{0.5}{V} \times m$

**Figure 5.** Options included in the students' task of research lesson 1 for students to select the equation that relates the kinetic energy (E) from the mass (m) and the speed (V) (translated from Portuguese in the task provided to the students).

To support the students in answering this question, the PSTs elaborated other representations also presented in the task statement. In concrete, these representations included the table with the values of kinetic energy (E) and the speed of a car (V) (Figure 3), and two graphs, one plotting the kinetic energy as a function of the mass, at constant speed, and the other graph plotting the kinetic energy vs. the speed of the car, at constant mass (Figure 6). The students tried to identify the correct algebraic equation to answer the question presented in Figure 5 using a trial-and-error approach and they had no success.

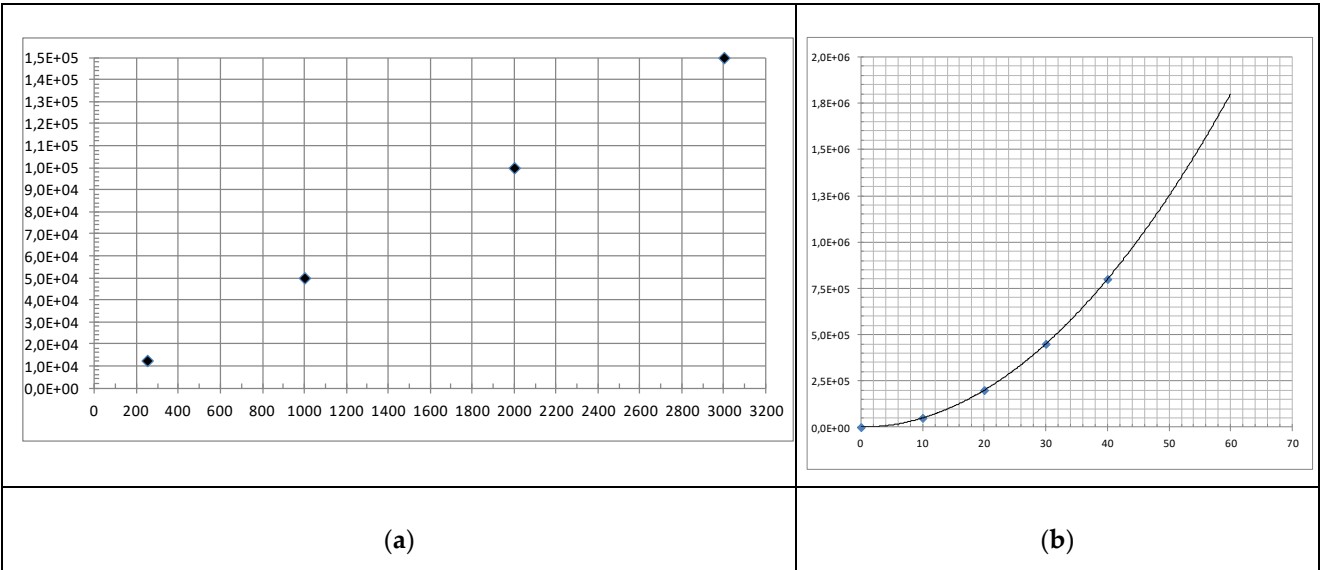

(**a**)

(**b**)

**Figure 6.** Graphs included in the task statement of research lesson 1 to support students in selecting the equation that correctly relates the kinetic energy (E) with the mass (m) and the speed (V): (**a**) kinetic energy vs. mass at constant speed and (**b**) kinetic energy vs. speed at constant mass (translated from Portuguese in the task provided to the students).

During the reflection session of research lesson 1, (session 13), when the PSTs were elaborating about the students' rationale to try to answer the question shown in Figure 5, there was the following dialogue:

PST3: The students picked some values from the tables and the graphs and made numerical substitutions in the algebraic equations to try to verify which one was correct. They did the calculations wrong, but the reasoning was correct.

PST1: This [student] also tried to answer the question [Figure 5] using the same approach, but since he used the values in the wrong units, his calculation was also incorrect (research lesson 1).

This transcript exemplifies an episode that gave the PSTs insights into students' difficulties in understanding the kinetic energy concept and that led to the results of Category 2 "Changes in the MR that PSTs use to support the teaching of the scientific concepts" (Table 3), as detailed below.

In fact, using abstract representations such as algebraic equations (Figure 5) and Cartesian graphs (Figure 6), the PSTs were not able to develop students' understanding of the kinetic energy phenomenon. Moreover, the use of algebraic equations to explore the relationships between the three scientific concepts, mass, speed, and kinetic energy (Figure 5), and the plotting of kinetic energy vs. speed at constant mass (Figure 6) to explore the relationships between the two concepts, speed and kinetic energy (Figure 3), reinforced the idea that mathematical models determine nature and not the reverse. Beyond this, neither the data included in the table (e.g., Figure 3) nor in the graphs (Figure 6) represented actual values of the kinetic energy because they were not experimental results, therefore they do not represent the real world from the 9th grade students' perspective.

In brief, in research lesson 1, the PSTs used complex and abstract representations, distant from the real world. This approach was not successful in supporting students to integrate the abstraction of scientific concepts regarding the kinetic energy.

Later, during reflection 1 (Figure 2), the PSTs designed solutions to meet students' needs in understanding the scientific concepts and the relationships among the variables involved. For instance, in reflection 1, PST2 and PST3 became aware of the importance of students observing an experience and visualizing (actual) data to better understand the scientific concepts as illustrated by the following dialogue between PST2 and PST3:

[02:11:47] PST2: If students do not collect the data, they will hardly understand the kinetic energy. This is a problem that has not yet been resolved.

[02:12:03] PST3: But we do not have time to change everything [task statement and lesson plan]!

[02:12:07] PST2: Right, but we can talk about the experimental data and the sensors used to collect them. Discussing this in class is a step forward.

[02:12:16] PST3: OK. I agree with you (Reflection 1).

This episode exemplifies the reasoning of PSTs that encouraged the changes in the way they came to represent the scientific concepts. In addition, the reference to the sensors for obtaining experimental data from an event shows the increased awareness of the PSTs about the importance of supporting their explanation on data from the "real" world. This solution was included in research lesson 2 (Figure 4).

The next episode, transcribed below, was extracted from the interview made to PST1. Similar description was made by PST3 in his interview.

[01:22:43] In the PST2 lesson [research lesson 2], we talked about the sensors and where they should be placed during the experiment [to collect data]. This discussion was very much participated in by the students and originated a conversation about sensors and what they measure. The students did not collect the data but understood how they are collected and then transformed to compute the maximum acceleration and force during the collision (PST1, Interview).

In fact, during research lesson 2, PST2 promoted a much participated-in discussion about sensors and what they measure, about the possible transformations of the data measured by the sensors, as well as how these data support the understanding of scientific concepts. Thus, from research lesson 1 to research lesson 2, the PSTs improved the representation of scientific concepts since they also used experimental data collected from a scientific article rather than only algebraic equations and other complex representations, such as the Cartesian graphs with fictional data used in research lesson 1. For this reason, this evidence constitutes a result of Category 2 (Table 3).

Results of Category 2 (Table 3) were also observed from cycle 2 to cycle 3 of the LS (Figure 2), as illustrated by the following episode. During reflection 2, the PSTs analyzed the students' responses and performance in carrying out the task that was proposed to them during research lesson 2 (the PSTs based these analyses on the students' responses to the task and in the video record of the lesson) and verified that some students continued to use scientific concepts in an abstract way, as well as continued not assigning a physical meaning to the concepts represented in a graph of acceleration over time. These PSTs' analyses catalyzed their design of different MR strategies of the car crash test to be implemented in research lesson 3 (taught by PST3) to help students to understand the scientific concepts (e.g., maximum acceleration, maximum force). As such, the data of the car crash test was shared with the students using a graph of the acceleration over time and simultaneously photos that represented different phases of the graph. Thus, in research lesson 3, the students also observed photos that were representations of events closer to real observable situations, and so the interpretation of the graph (which is a more abstract representation) became easier. The following dialogue occurred during reflection 2 (cycle 2, Figure 2) when PST2 was planning to change the representation of the event and illustrates a result of Category 2:

[01:52:09] PST3: ( . . . ) Students have many difficulties in mathematics. They will not be able to interpret the graphs.

[01:52:14] PST1: But we can start research lesson 3 with images showing the position [of the head of the dummy used in car crash-tests] at different moments of collision on the wall crash-test and then present the relationship between the images and the corresponding points on the acceleration graph [over time]. This will help a lot the graph interpretation (reflection 2).

In fact, during the enactment of research lesson 3, PST3 encouraged students to discuss the different representations of the crash-test shown and their relation with the scientific concepts using a sequence of photos of the dummy head in different instants of the collision, and simultaneously a Cartesian graph plotting a sensor output of the acceleration of the dummy head over time during the collision. PST3 asked the students the following question "What is the required information to evaluate the safest baby seat model?" (field notes from research lesson 3). After 7 min of an extremely rich and intense discussion between PST3 and the students about the information needed to answer the question, PST3 wrote on the whiteboard with students' help the following expressions: "Car seat model/Price/Maximum acceleration on the dummy head $(m/s^2)$/Maximum force on the dummy head (N)" (field notes from research lesson 3). The dialogue between PST3 and the students proceeded as follows:

[00:23:05] PST3: To evaluate the safest baby seat model we need to collect data regarding each of the items written in the whiteboard [points to the whiteboard]. As a group, you need to plan a process for organizing and collection the data. Let's do this!

[00:23:17] Student 4: What data?

[00:23:19] PST3: What data do we need to evaluate the safest baby seat model and how can you collect it?

[00:23:27] Student 13: [We need to know] the price of each model, the acceleration, and the strength. You have already said which are the models [of baby chairs].

[00:23:26] PST3: The acceleration of what? And the applied force in what? We are talking about objects, thus we need to identify those objects when explaining the ideas (research lesson 3).

This discussion illustrates, again, changes verified in PST3 regarding the use of the MR of the scientific concepts that occurred from research lesson 2 to research lesson 3, namely the word "data" emerged several times during this dialogue with the students. This supported students in employing representations closer to the real world, i.e., more "real" and "concrete" representations than, for instance Cartesian graphics. Consequently, the students were in better conditions to understand the scientific concepts.

Moreover, PST3 assisted students to reconstruct the experience (i.e., the position of the dummy head during a crash-test) in which data was measured with sensors and explained the experiment including the scientific concepts, such as the acceleration using scientific actual data obtained in the literature. Consequently, the students explored the real world while appropriating the scientific concepts (e.g., acceleration and strength applied on the dummy head) and constructed their own meanings.

In addition, PST3 chose not to organize the data related with the crash test experiment, in particular the mass of the dummy head as well as its maximum force and acceleration (collected with the sensor during the collision) as well as other relevant information (such as the models of the baby chairs and price). Instead, PST3 encouraged the students to elaborate their own organization of the experimental data and then to transform them into other representations. In this way, PST3 was able to observe the students' reasoning and logic, helping them to mitigate the abstraction of the scientific concepts through their relation with the concrete application under analysis. In fact, the experimental data and its transformation are very important representations for understanding the concepts of science.

Thus, the group analysis of the students' results provided insights about the students' difficulties and showed to be adequate since it triggered changes in the PSTs on the MR of

the scientific concepts regarding kinetic energy. The aforementioned changes on the use of MR were intentionally planned and executed by the PSTs and occurred from research lesson 1 to research lesson 2 (cycle 1 to cycle 2 of the LS) as well as from research lesson 2 to research lesson 3 (cycle 2 to cycle 3).

In summary, regarding "Changes in the MR that PSTs use to support the teaching of the scientific concepts", from research lesson 1 to research lesson 2 (i.e., from the end of cycle 1 to cycle 2), the PSTs improved the representations to support the teaching of the scientific concepts since they used a sensor output of the acceleration of the dummy head in different instants of the collision to explore the relationship between variables rather than with fictional data, and algebraic equations and graphs both distant from the experience (car crash test).

From research lesson 2 to research lesson 3, PST3 became aware of the importance of students observing an experience and visualizing a car crash test. As such, the PSTs again improved the representations to support the teaching of the scientific concepts since they used a sequence of photos of the dummy head in different instants of the collision and a table elaborated by the students to explore variables and its relations and transformations. Furthermore, in research lesson 3, the (actual) "data" (a representation) emerged several times during this dialogue with the students. This supported students in employing representations closer to the real world, i.e., more "real" and "concrete" representations than graphs and algebraic equations.

## 5. Discussion

The aim of this research was to investigate changes, promoted by the LS with three cycles, in the PSTs' practices regarding the use of MR in teaching kinetic energy. During the LS sessions, the PSTs were encouraged to deepen their knowledge on the use of MR as well as to prepare, teach, and improve their practice applying MR. The support for PSTs by the teacher educators and researchers was guided through inquiry-based strategies (see 3.2. Lesson Study Implementation section).

To examine the changes (and their effect) on PSTs practices, both students and PSTs activities and performance were collected in 51 h of video records from all sessions of the LS, in 5.5 h of interviews to the PSTs, and in field notes taken in all LS sessions (by the three authors). In addition, the materials produced by the PSTs, namely, the research lesson plans and the tasks prepared for the students to do during the lessons, were analyzed. All these data showed the occurrence of changes in the PSTs practice over time. From data analyses of the material collected iteratively emerged two categories of analysis; category 1, changes in the MR that PSTs used to represent cars crash-tests as the "real" world event, and category 2, changes in the MR that PSTs used to support the teaching of the scientific concepts (Table 3).

It is important to discuss which activities the PSTs conducted and among these, which were the ones that most likely triggered the occurrence of changes in the PSTs' practice.

The activities carried out by the PSTs between the enactment of two subsequent research lessons (i.e., between research lesson 1 and research lesson 2 and between research lesson 2 and research lesson 3, Figure 2) were basically the same: teaching with observation, and the research lesson supported by a robust and coherent lesson plan.

Analyzing the material collected during the previous research lesson, namely, both students' responses to the task and video record segments, as well as defining strategies to use MR in the following research lesson aimed to provide an improved support for students' progress in their learning. These activities were performed during the corresponding reflection sessions.

Teaching with observation the following research lesson included the refined strategies regarding the use of MR in teaching the kinetic energy topic.

Thus, the activities from research lesson 1 to research lesson 2 (i.e., cycle 1 to cycle 2) and from research lesson 2 to research lesson 3 (i.e., cycle 2 to cycle 3)—namely, teaching and observing, collecting student data, investigating challenges to student learning, creating

possible solutions to help students learning and (re)testing—were the activities of the LS that triggered the occurrence of changes regarding the use of MR in the PSTs' practice. Since there were changes over the three cycles, (i.e., from cycle 1 to 2 and from cycle 2 to 3) (Figure 2), one may infer that the three cycles of the LS had a positive effect on the results obtained in this investigation. However, from research lesson 1 to research lesson 2 (i.e., end of the cycle 1 to cycle 2), changes occurred both in the MR of the event (category 1) (see Section 4.1) and in the MR of the teaching of scientific concepts (category 2) (see Section 4.2) while from cycle 2 to cycle 3 only occurred changes in the MR of the teaching of scientific concepts (see Section 4.2). That occurred because from cycle 2 to cycle 3, the MR of the event (category 1) remained successful (see Section 4.1, last paragraph).

Previous research studies seemed to encourage the implementation of more than one cycle in the LS, in the context of PSTs teacher education [18,20,21,41,52,53]. Similar results were found in the current investigation. Despite these promising results concerning the repetition of LS cycles, researchers [38] argued that in case of in-service teacher education, the group of teachers rarely repeats a cycle in LS in Japan. Researchers [40] showed that outside of Japan, the effectiveness of LS is uneven due to misunderstandings on the structure and certain practices of this teacher education program. Namely, outside of Japan, it is common to omit the lesson planning that helps teachers to gain knowledge and insight into science as well as about student thinking [47]; to fit an entire LS cycle into one day [48]; and to repeat the research lesson six times.

The purpose of the present study was to use the LS to gain new knowledge for teaching and learning rather than to create a perfect lesson plan. Thus, this investigation included three sessions of three hours each to perform lesson planning (Figure 2) as part of long-term LS (17 sessions). In addition, the goal of implementing three research lessons was to provide opportunity for each PST to teach and observe a research lesson instead of obtaining a "perfect" lesson plan. As such, this investigation intended to follow the original Japanese lesson study [40] as much as possible, as it had to be adjusted to the PSTs' teacher education.

In other teacher education contexts, the implementation of LS also showed that the use of MR by PSTs improved, in particular, the use of more diverse and complex representations over time [8]. In addition, previous studies on the use of MR [6] suggested that PSTs require involvement in activities that allow reflection upon practice, critique by other participants, instructional revision strategies, and reimplementation, as typically occurs in LS and as it was followed in the current investigation (see Section 3.2, Lesson Study Implementation). Thus, it might be inferred that these were the activities that triggered the occurrence of changes in the PSTs' practice.

The current investigation was performed using the LS with an adaptation (namely, the adjustment of the LS sessions' content to the previous PSTs' knowledge) and involving a small number of PSTs participants (n = 3). In addition, the LS implementation used a specific topic (in concrete, the kinetic energy) that is taught to 9th grade students. Thus, the findings of this investigation are valid for the present results. The conclusions' generalization to a wider range of scientific topics (physics and chemistry) and for all PSTs may be, partially or totally, invalid.

It remains for future research studies to provide a deeper understanding of what other elements of the LS with PSTs may contribute to enhance their learning regarding MR, and to other fields related to teachers' professional knowledge. Studies with PSTs in LS using a larger number of groups are to be pursued in future investigations.

## 6. Conclusions

This investigation showed that from cycle to cycle in the LS with PSTs, the MR of the car crash test event well as the MR of the scientific concepts for teaching kinetic energy became closer of an observable event that could happen in the "real" world. At the beginning of the LS, the PSTs used in their practice more abstract and complex representations. However, while the LS implementation unfolded, the PSTs also started to include more concrete

representations to facilitate student learning. The findings give more insight and deepens knowledge regarding the use of MR in physics with PSTs in two perspectives: in the science educators' perspective, since it shows how PSTs use MR in physics teaching and improvements over time, as well as the feasibility of including the LS approach in initial teacher programs, and from the researchers' perspective, since it concurs to increase the literature about the use of MR in science with PSTs in LS.

**Author Contributions:** Conceptualization, T.C., M.B. and J.P.P.; methodology, T.C., M.B. and J.P.P.; validation, J.P.P.; formal analysis, T.C. and M.B.; investigation, T.C., M.B. and J.P.P.; data curation, T.C., M.B. and J.P.P.; writing—original draft preparation, T.C.; writing—review and editing, T.C., M.B. and J.P.P.; visualization, T.C.; supervision, M.B.; project administration, J.P.P.; funding acquisition, T.C. All authors have read and agreed to the published version of the manuscript.

**Funding:** This research was funded by Fundação para a Ciência e a Tecnologia (FCT) under Grant SFRH/BD/147648/2019.

**Institutional Review Board Statement:** The study was conducted according to the guidelines of the Declaration of Helsinki, and approved by the Ethics Committee of Instituto de Educação da Universidade de Lisboa (date of approval: 15 February 2021).

**Informed Consent Statement:** Informed consent was obtained from all subjects involved in the study.

**Data Availability Statement:** The data presented in this study are available on request from the corresponding author.

**Conflicts of Interest:** The authors declare no conflict of interest.

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
