# Peer review of "Lesson Study as a Means to Change Secondary Preservice Physics Teachers’ Practice in the Use of Multiple Representations in Teaching"

_education, doi:10.3390/educsci11120791_

Round 1

Reviewer 1 Report

The article analyzes a topic of interest and focused on the focus of the journal.

The design and methodology used are adequate to achieve the proposed objectives.

Although the study has few participants, I think the article may be of interest.

Here are some suggestions that I hope will be helpful to authors:

1. Bibliographic references.
The bibliographic references cited must be updated. Only 12 of the 48 total references have been published in 2016 or later.

2. Study limitations.
It is recommended to expose the limitations of the study at the end of the discussion.
Additionally, it is recommended to relocate future research lines also at the end of the discussion (instead of in the conclusions section).

3. Formal issues.
3.1. The bibliographic references do not follow the norms of the journal.

3.2. There is a formatting problem in Table 1 and the text that precedes it.

3.3. It is recommended to modify the format of figure 3 so that it is better visualized.

Author Response

AUTHORS’ RESPONSE TO REVIEWER 1

Lesson study as a mean to change secondary preservice physics teachers’ practice for the use of multiple representations in teaching

The authors would like to thank the time and constructive criticisms of the Reviewers of this manuscript that strongly contributed to its improvement.

The authors have addressed each of the Reviewers queries/comments as outlined in Table 1. In this table, the reviewer’ s comments and suggestions are presented with a grey background whereas the authors’ responses have white background. All the modifications to the original manuscript are in track changes.

Table 1 – Answers to Reviewer #1

Reviewer Comment: 1. Bibliographic references.
The bibliographic references cited must be updated. Only 12 of the 48 total references have been published in 2016 or later.

Authors’ response:

To implement the Reviewer suggestion, the bibliographic references were updated.

 [32] Arias, A. M.;  Davis, E. A. Supporting children to construct evidence-based claims in science: Individual learning trajectories in a practice-based program, Teaching and Teacher Education, 66, 204-218, 2017 doi:10.1016/j.tate.2017.04.011

[33] Bowen, G. M.; Bartley, A.; MacDonald, L..; Sherman, A. Experiences with Activities Developing Pre-service Science Teacher Data Literacy. In, Buck, G.; Akerson, V. Eds.; Enhancing Professional Knowledge of Pre-Service Science Teacher Education by Self-Study Research. ASTE Series in Science Education, pp 243-269, 2016, Springer, Cham. doi:10.1007/978-3-319-32447-0_13

[34] Enzingmüller, C.; Prechtl, H.; Constructing Graphs in Biology Class: Secondary Biology Teachers’ Beliefs, Motivation, and Self-Reported Practices, International Journal of Science and Mathematics Education, 19, 1-19, 2021. doi:10.1007/s10763-019-09975-2

[35] Clark-Wilson, A.; Hoyles, C. A. Research-informed web-based professional development toolkit to support technology-enhanced mathematics teaching at scale. Educ Stud Math 102, 343–359, 2019. https://doi.org/10.1007/s10649-018-9836-1

[36] Gardner, S.; Suazo-Flores, E.; Maruca, S.; Abraham, J.; Karippadath, A.; Meir, E. Biology Undergraduate Students' Graphing Practice in Digital Versus Pen and Paper Graphing Environments. Journal of Science Education and Technology, 30, 431-446, 2021. doi:10.1007/s10956-020-09886-w.

Reviewer Comment: 2. Study limitations.
It is recommended to expose the limitations of the study at the end of the discussion.
Additionally, it is recommended to relocate future research lines also at the end of the discussion (instead of in the conclusions section).

Authors’ response:

To implement this Reviewer’s suggestion, the limitations of the study and future research are now at the end of the discussion (instead of in the conclusions section). As such the amended text are currently on page 15, after the last paragraph of the “Discussion” section, as follow:

·         The current investigation was performed using the LS with an adaptation (namely, the adjustment of the LS sessions’ content to the previous PSTs’ knowledge) and involving a small number of PSTs participants (n=3). In addition, the LS implementation used a specific topic (in concrete, the kinetic energy) that is taught to 9th grade students. Thus, the findings of this investigation are valid for the present results. The conclusions’ generalization to a wider range of scientific topics (physics and chemistry) and for all PSTs may be, partially or totally, invalid.

It remained for future research studies to provide a deeper understanding of what other elements of the LS with PSTs may contribute to enhance their learning regarding MR, and to other fields related to teachers’ professional knowledge. Studies with preservice teachers in LS using a larger number of groups are to be pursued in future investigations.

Reviewer Comment: 3. Formal issues.
3.1. The bibliographic references do not follow the norms of the journal.

3.2. There is a formatting problem in Table 1 and the text that precedes it.

3.3. It is recommended to modify the format of figure 3 so that it is better visualized.

Authors’ response:

Taking into consideration the Reviewer’s suggestions:

3.1. The bibliographic references were amended and are now accordingly the norms of the journal

3.2. The Authors agree with the Reviewer’s remark. Thus the text that precedes the Table 1 is now outside of the Table 1

3.3. The Authors agrees with the Reviewer’s remark. Although the image was extracted from the video recording. As such unfortunately the Authors are not able to improve the quality of the format of Figure 3. Although the size of the figure 3 is now bigger so that it is better visualized.

Reviewer 2 Report

General Comments and Suggestions

  1. The references are not following the MDPI format.
  2. There is a need to describe the representation of the phenomenon that was used in the research lesson 2 in order to compare it with the representation used in RL 1.  Consider also to justify the relevance of these representations with a car crash test, which is the main topic of the lessons.
  3. In the Results section it would be helpful if the authors made clearer the important changes that the PSTs made from one cycle to the other. The way the changes are presented now, is a little bit confusing, as the authors discuss the changes and the data that lead to the changes altogether.
  4. The discussion and conclusion sections comprise mostly a description of the results than a discussion in relation to the literature. So, the authors should minimize the parts that are description of the results, as this information is already presented and strengthen the parts that provide explanations of the results and teaching implications as a result of the study’s results.

Comments and suggestions by Line

Line 21: What is the first image named “Aim” just before the Introduction? To my knowledge the paper should begin with the Introduction and not with an image as a synopsis.

Line 21-45 or/and 64-104: There is a need for a better definition of what the authors mean with the term Multiple Representations. Multiple Representations of what? For instance, are they the product of a simulation or the simulation itself? An illustration, a table or a graph are mostly products of a procedure, and they can be representations of a concept/phenomenon. On the other hand, a web-based simulation itself could be used as a means for the students to express their view about a concept/phenomenon. I have the sense that the authors use the term MR for both the product and the tool for representing a phenomenon. This produces a slight confusion in the manuscript. This clarification could be done somewhere in lines 21-45, or /and 54-104adding a small or two sentences.

Line 30: Why XX century and not 20th century?

Line 30-38: An emphasis on the fact that using MR would help students and teachers understanding because each representation focus on a different aspect of the concept or the phenomenon is not apparent in the manuscript. This could be mentioned in the second paragraph of the introduction.

Line 55: Reference (Larssen et al., 2018; Author, 2017) is in blue. Maybe the phrase “teacher educator support” needs restatement in order to be clear who needs support- the teacher educator or teachers by the educator?

Line 57: It is mentioned that “The knowledge on the LS applicability to develop PSTs’ ability to use of MR in physics teaching is still scarce”. It would be useful to refer to other research findings e.g. reviews confirming this claim.

Line 70: The phrase “that might be used in science teaching by combination or superimposed” needs clarification or verbal restatement e.g.

Lines 83-104: These twenty lines describe two teacher education programs (Bowen & Roth, 2005; Lunsford et al., 2007) aiming to promote use of MR in science teaching. The paper could benefit substantially  from a more extended and recent literature review on teacher education programs in promoting MR.

Line 153: Section 3.1. Characterization of the Participants in the Research.” This section needs proper format and language editing. Specifically, reference to other sections in this text needs attention. Also, it would be better for the main text “  The Participants in the research were the PSTs (n = 3) - PST1, PST2 and PST3 - enrolled in a teacher education course to obtained a certification to teach science (physics and chemistry) at lower and upper secondary level (see more information at Table 1). In the LS sessions (detailed in section 2.2) participated the PSTs (section 3.1), three cooperating teachers, a Professor of the University (second author) that offers the science teacher trainee and certification, two educational researchers (first and third authors) and a physics re-searcher. The option to involve cooperating teachers (e.g., Parks et al., 2020; Zeichner, 2010) as well as re-searchers (Lunsford et al., 2007) stemmed from recognized relevance of these participants in the PSTs train”, to be excluded from Table 1. Also, there seems to be a need for a language check in the specific text.

Line 155: The authors write “The activities of the PSTs took place at the university were described in Figure 2”. However, figure 2 does not describe any activities. It shows the sequence of cycles of the LS and the sessions in each cycle.

Lines 173-175: Cycle three needs to be rephrased in a more simple and comprehendible way. It seems that cycle 3 included a Research Lesson (session16) conducted by PST3, followed by a Reflection 3 session.

Lines 190-213: The methodological purpose leading the first researcher to analyze qualitative data into specific categories from all video records (sessions 1-17) while the second researcher analyzed also video records and field notes from sessions 9-17 it is not clear to the reader and needs to be explained. It is also mentioned that the categories were “refined according to the code redefinition in order to answer the research question” . Additionally more information and some indicative examples about the coding procedure (and its’ redefinition) would be most useful for the understanding of the methodology followed.

The purpose of the study is initially mentioned in the discussion (line 527, page 13). It would be much more easier for the reader to understand the research framework if the paper stated the purpose of the study and the research question explicitly in the Method section.

Page 7, content of “Table 2. Categories that emerged from data analysis” needs language checking

Line 194 and 199: Content analysis usually uses predefined categories while grounded theory seeks to define them through iterative comparison of the data. The authors should describe in a clearer manner how they combined these two approaches that they refer in lines 194 and 199 respectively.

Line 214: The difference between the two categories in Table 2 is not sufficiently clear neither in the method nor in the results section.

Lines 317 and 318: It should not be in the main text as they are a part of the task following figure 4.

Line 476: In the discussion section it is mentioned that the PSTs were encouraged to deepen their knowledge on the use of MR. More information about the way they were encouraged by the research team is needed, in order to understand the procedure that took place helping PSTs to readjust their teaching practices using MR.

Lines 515-534: Explanations about the way the Lesson Studies were implemented, in comparison to other research using the same or similar methodology could be mentioned in the method section, since the way they are connected to the discussion of results is not clear.

The authors need to make a more clear connection between result and cause in the discussion, justifying why from Research Lesson 1 to Research Lesson 2 (i.e., end of the cycle 1 to cycle 2) occurred changes both in the MR of the event (Category  1) and in the MR of the scientific concepts (Category 2) while from cycle 2 to cycle 3 only  occurred changes in the MR of the scientific concepts and possibly discuss it with other similar or contradicting research findings.

Author Response

AUTHORS’ RESPONSE TO REVIEWERS

Lesson study as a mean to change secondary preservice physics teachers’ practice for the use of multiple representations in teaching

The authors would like to thank the time and constructive criticisms of the Reviewers of this manuscript that strongly contributed to its improvement.

The authors have addressed each of the Reviewers queries/comments as outlined in Table 1. In this table, the reviewer’ s comments and suggestions are presented with a grey background whereas the authors’ responses have white background. All the modifications to the original manuscript are in track changes.

Table 1 – Answers to Reviewer #2

Reviewer Comment: 1. The references are not following the MDPI format.

Authors’ response:

Taking into consideration the Reviewer’s suggestion, the references are now following the MDPI format.

Reviewer Comment: 2.1. There is a need to describe the representation of the phenomenon that was used in the research lesson 2 in order to compare it with the representation used in RL 1. 

2.2. Consider also to justify the relevance of these representations with a car crash test, which is the main topic of the lessons.

Authors’ response:

2.1. The Authors agree with Reviewer’s suggestion. As such more information was added to the manuscript, on page 9, between fourth and fifth paragraph, as follows:

·         Figure 4 presents an example of realistic crash-tests included in the students’ task of the Research Lesson 2 for two different models of baby chairs:

Figure 4 Realistic crash-tests included in the students’ task of the Research Lesson 2 for two different models of baby chairs (in English language “Aceleração” is “Acceleration”, “Tempo” is “Time” and “Modelo” is “Model”).

2.2. The Authors thank you the opportunity to improve such issue. Thus, more information was added to the manuscript, currently on page 3 at the end of the “Introduction” section, as follows:

·         This paper aims to make a contribution in this regard. Thus, the aim of this research is to examine changes in the practices of a group of PSTs regarding the use of MR in the teaching of the kinetic energy at 9th grade. For this purpose, a LS with three cycles was implemented based on the car crash-test as the “real” world event. A car crash-test is a real life situation and a common phenomenon that helps students linking abstract concepts with concrete phenomena and enables them to better conceptualize scientific concepts [9].

Reviewer Comment: 3. In the Results section it would be helpful if the authors made clearer the important changes that the PSTs made from one cycle to the other. The way the changes are presented now, is a little bit confusing, as the authors discuss the changes and the data that lead to the changes altogether.

Authors’ response:

In order to clearer changes that the PSTs made from one cycle to the other more information was added to the revised manuscript, on page 13, after the last paragraph of the “Results” section, as follows:

·         In summary, about “Changes on the MR that PSTs use to support the teaching of the scientific concepts”, from Research Lesson 1 to Research Lesson 2 (i.e., from the end of cycle 1 to cycle 2), the PSTs improved the representations to support the teaching of the scientific concepts since they used a sensor output of the acceleration of the dummy head in different instants of the collision to explore the relationship between variables rather than with fictional data, and algebraic equations and graphs both distant from the experience (crash car-test).

From Research Lesson 2 to Research Lesson 3, PST3 became aware of the importance of students observing an experience and visualise a car crash-test. As such, the PSTs improved again the representations to support the teaching of the scientific concepts since they used a sequence of photos of the dummy head in different instants of the collision, and a Table elaborated by the students, to explore variables and its relations and transformations. Furthermore, in Research Lesson 3, the word (actual) “data” (a representation) emerged several times during this dialogue with the students. This supported students to employ representations closer to the real world, i.e., more “real” and “concrete” representations than graphs and algebraic equations.

Reviewer Comment: 4. The discussion and conclusion sections comprise mostly a description of the results than a discussion in relation to the literature. So, the authors should minimize the parts that are description of the results, as this information is already presented and strengthen the parts that provide explanations of the results and teaching implications as a result of the study’s results.

Authors’ response:

Accordingly the Reviewer’s remark some description of the results on page 14 on the “Discussion” section was removed from the manuscript, after the second paragraph. These modifications are in track changes on the revised manuscript.

Reviewer Comment: Line 21: What is the first image named “Aim” just before the Introduction? To my knowledge the paper should begin with the Introduction and not with an image as a synopsis.

Authors’ response:

The Authors thank you the opportunity to improve such question. The first figure in the manuscript is the Graphical Abstract that is typically placed by the editors at the top of the manuscript (just before the introduction and just after the abstract in text). The Graphical Abstract was designed to be a unique, compact, and visual summary of the all main aspects of the manuscript. The main intention is to capture the attention and interest of the reader to the manuscript content in an easy and fast way.

The word “Aim” to which the Reviewer alludes belongs to the Graphical Abstract of the present manuscript. In order to postpone such confusion, the Authors introduced the title, “Graphical Abstract” on the top of the figure in the manuscript.

Reviewer Comment: Line 21-45 or/and 64-104: There is a need for a better definition of what the authors mean with the term Multiple Representations. Multiple Representations of what? For instance, are they the product of a simulation or the simulation itself? An illustration, a table or a graph are mostly products of a procedure, and they can be representations of a concept/phenomenon. On the other hand, a web-based simulation itself could be used as a means for the students to express their view about a concept/phenomenon. I have the sense that the authors use the term MR for both the product and the tool for representing a phenomenon. This produces a slight confusion in the manuscript. This clarification could be done somewhere in lines 21-45, or /and 54-104adding a small or two sentences.

Authors’ response:

The Authors agree with Reviewer’s suggestion. Thus the text was amended on page 3, of the “Background” section, “2.1. Multiple Representations in Physics Teaching” subsection, at the end of the first paragraph, as follow:

·           These interactions include production, reading, transformation, and evaluation of MR [3]. In the scope of this study, MR means a product of a procedure (such as, table and a graph) and the tool for representing a phenomenon.  Furthermore, constructing MR actively engages students in their learning and develops their thinking, predicting, making claims, understanding and representing skills [29- 31].

Reviewer Comment: Line 30: Why XX century and not 20th century?

Authors’ response:

The Authors agree with the Reviewer´s remark. As such instead of “XX century” in the “Introduction” section, second paragraph, the Authors wrote “20th century”

Reviewer Comment: Line 30-38: An emphasis on the fact that using MR would help students and teachers understanding because each representation focus on a different aspect of the concept or the phenomenon is not apparent in the manuscript. This could be mentioned in the second paragraph of the introduction.

Authors’ response:

The Authors agree with the Reviewer’s remark. As such more information was added to the manuscript in the second paragraph of the “Introduction” section, as follows:

·         MR have been under intense scrutiny in the field of education since the end of the 20th century [e.g., 2, 5-8]. Several studies have shown that the use of MR in science teaching enables students to better conceptualize scientific concepts as they can make sense of real life situations and common phenomena [3, 5, 9]. In addition, MR promote students’ interest in learning scientific concepts by providing opportunities for linking abstract concepts with concrete phenomena [10, 11]. Beside that using MR would help students and teachers better understanding concept or the phenomenon because each representation focuses on a different aspect of the concept or the phenomenon. Therefore, they are an excellent resource for students to learn science concepts [3, 5].

Reviewer Comment: Line 55: Reference (Larssen et al., 2018; Author, 2017) is in blue. Maybe the phrase “teacher educator support” needs restatement in order to be clear who needs support- the teacher educator or teachers by the educator?

Authors’ response:

The Authors agree with the Reviewer’s question. Thus the text in the manuscript was amended as follows:

·         This approach involves teachers in constant questioning about teaching practice in order to improve it. Previous researches in LS showed promising results as a learning process of PSTs [e.g., 18-22] although requiring support by the teacher educators [23, 24].

Reviewer Comment: Line 57: It is mentioned that “The knowledge on the LS applicability to develop PSTs’ ability to use of MR in physics teaching is still scarce”. It would be useful to refer to other research findings e.g. reviews confirming this claim.

Authors’ response:

The Authors agree with Reviewer suggestion. Thus a revision literature was made and the following text was added to the revised manuscript, as follows:

·         Indeed, from the results of a revision literature on web of sciences and Scopus with the words “lesson study”, “representations”, “graphs”, “tables”, “simulations” “physics” only appears the [44] research

Reviewer Comment: Line 70: The phrase “that might be used in science teaching by combination or superimposed” needs clarification or verbal restatement e.g.

Authors’ response:

The Authors added more information to clarify the phrase. Thus the following text in the manuscript was added:

·         “that might be used in science teaching by combination i.e., using various representations separately or superimposed, i.e., laid over other representations” [27].

Reviewer Comment: Lines 83-104: These twenty lines describe two teacher education programs (Bowen & Roth, 2005; Lunsford et al., 2007) aiming to promote use of MR in science teaching. The paper could benefit substantially  from a more extended and recent literature review on teacher education programs in promoting MR.

Authors’ response:

Accordingly the Reviewer’s suggestion more recent literature was added to the manuscript, on page 3 of the “2.1. Multiple Representations in Physics Teaching” section, second paragraph, as follows:

·         In fact, research showed the importance of including MR training in initial teacher education programs to support its use by PSTs in teaching science [32-36]

Reviewer Comment: i) Line 153: Section 3.1Characterization of the Participants in the Research.” This section needs proper format and language editing. Specifically, reference to other sections in this text needs attention.

ii) Also, it would be better for the main text “  The Participants in the research were the PSTs (n = 3) - PST1, PST2 and PST3 - enrolled in a teacher education course to obtained a certification to teach science (physics and chemistry) at lower and upper secondary level (see more information at Table 1). In the LS sessions (detailed in section 2.2) participated the PSTs (section 3.1), three cooperating teachers, a Professor of the University (second author) that offers the science teacher trainee and certification, two educational researchers (first and third authors) and a physics re-searcher. The option to involve cooperating teachers (e.g., Parks et al., 2020; Zeichner, 2010) as well as re-searchers (Lunsford et al., 2007) stemmed from recognized relevance of these participants in the PSTs train”, to be excluded from Table 1.

iii) Also, there seems to be a need for a language check in the specific text.

Authors’ response:

i) The Authors agree with Reviewer’s remark. As such the text in the manuscript, on page 4, of the “3.1. Characterization of the Participants of the Research” section, first paragraph, was amended, as follows:

·         The Participants in the research were the PSTs (n = 3) - PST1, PST2 and PST3 - enrolled in a teacher education course to obtained a certification to teach science (physics and chemistry) at lower and upper secondary level (see more information at Table 1). In the LS sessions (detailed in section 3.2) participated the PSTs, three cooperating teachers, a Professor of the University (second author) that offers the science teacher trainee and certification, two educational researchers (first and third authors) and a physics researcher. The option to involve cooperating teachers [e.g., 45, 46] as well as researchers [8] stemmed from recognized relevance of these participants in the PSTs training.

ii) Accordingly with Reviewer’s suggestion the text mentioned by the Reviewer was removed from Table 1

iii) The authors check the language.

The participants in the research were the PSTs (n = 3) - PST1, PST2 and PST3 - enrolled in a master degree in teaching science (physics and chemistry) at lower and upper secondary level (see more information at Table 1). In the LS sessions (detailed in section 3.2) participated the three PSTs, three cooperating teachers, one Professor of the University (second author) that conducted the LS sessions, two educational researchers (first and third authors) and one physics researcher. The option to involve cooperating teachers [e.g., 45, 46] as well as researchers [8] stemmed from recognized relevance of these participants in the PSTs training.

Reviewer Comment: Line 155: The authors write “The activities of the PSTs took place at the university were described in Figure 2”. However, figure 2 does not describe any activities. It shows the sequence of cycles of the LS and the sessions in each cycle.

Authors’ response:

The Authors agree with Reviewer’s remark. As such the phrase referred by the Reviewer “The activities of the PSTs took place at the university were described in Figure 2” was amended as follow:

·         The sessions of the LS with the PSTs took place at the university and were schematically represented in Figure 2

Reviewer Comment: Lines 173-175: Cycle three needs to be rephrased in a more simple and comprehendible way. It seems that cycle 3 included a Research Lesson (session16) conducted by PST3, followed by a Reflection 3 session

Authors’ response:

The Authors clearer the description of the ‘Cycle 3’, as follows:

·         ‘Cycle 3’ comprised the two sessions that reproduced those of the ‘Cycle 2’ i.e., Research Lesson 3 (session 16) and Reflection 3 (session 17), except that the Research Lesson 3 (session 16) was taught by the third pre-service teacher (PST3).

Reviewer Comment: 1.1) Lines 190-213: The methodological purpose leading the first researcher to analyze qualitative data into specific categories from all video records (sessions 1-17) while the second researcher analyzed also video records and field notes from sessions 9-17 it is not clear to the reader and needs to be explained.

1.2) It is also mentioned that the categories were “refined according to the code redefinition in order to answer the research question”. Additionally more information and some indicative examples about the coding procedure (and its’ redefinition) would be most useful for the understanding of the methodology followed.

1.3) The purpose of the study is initially mentioned in the discussion (line 527, page 13). It would be much easier for the reader to understand the research framework if the paper stated the purpose of the study and the research question explicitly in the Method section.

1.4) Page 7, content of “Table 2. Categories that emerged from data analysis” needs language checking

Authors’ response:

The Authors agree that the option for one of the authors analysing all data instead from session 9 to17, period in which the PSTs began to work in the teaching practice needs a justification. As such the text of the manuscript, on page 7, of the “3.3. Data Collection and Analysis” section, second paragraph, was amended as follows:

1.1) The video records of sessions 1 to 17 (about 51 hours in total) and the interviews (about five hours in total), were transcribed, and together with field notes of all sessions were analyzed to find changes that occurred in the PSTs’ practices regarding the use of MR. For this purpose, a descriptive and content analysis [47] was performed for qualitative data. In particular, the transcripts of video records and field notes were read by the first author of this article. After that, the targeted text was segmented, representing ideas related with the research aim. Each segment was assigned a code and clustered on a category, according to its features. The codes were refined through a constant questioning and comparison of the data [48]. The categories were also refined according the codes redefinition until they answer to the research question. Next, the second author analysed the transcripts of the video records of sessions 9 to 17 and field notes of the correspondent sessions. Indeed, the PSTs only worked in the teaching practice about the kinetic energy in sessions 9 to 17. The option of the first author to analyse the data from sessions 1 to 8 was due to the fact that the identification of changes in the practices of the PSTs with the MR (research objective) requires, in various situations, to understand what the PSTs already knew during the preparatory study (Session 1-8).

The Authors agree with Reviewer’s remark 1.2). As such the text of the manuscript, on page 7 of the “3.3. Data Collection and Analysis” section, third paragraph, was amended as follows:

1.2) Next, the second author analysed the transcripts of the video records of sessions 9 to 17 and field notes of the correspondent sessions. After that, based on the two descriptive categories that emerged from the data analysis, the second author analysed the codes, redefining the codes and created other codes if necessary. For example for one of the categories, one of the codes was “Using the MR to support the teaching of the scientific concepts”. This code was successively redefined as follows: First, “Using the MR to visualize the kinetic energy”, secondly, “Using the MR for the explanation of the kinetic energy”, third, “Using the MR of the scientific concepts”, and fourth, “Using the MR for the understanding of the scientific concepts”. The final code was “Using the MR to support the teaching of the scientific concepts” as already mentioned. This redefinition of codes resulted from the analysis and comparison of data iteratively carried out by the first two authors. The two researchers compared their codes and discussed a consensus coding scheme, which was 89 %. [49] proposal was considered for the calculation of reliability.

The Authors agree with the Reviewer’s comment. As such the purpose of the study and the research question are in the Method in concrete in “3.3 Data Collection and Analysis” section, page 7, first paragraph, in the revised manuscript as follow:

1.3) The aim of this research is to examine changes in the practices of a group of PSTs regarding the use of MR in the teaching of the kinetic energy at 9th grade. For this purpose, a LS with three cycles was implemented based on the car crash-test as the “real” world event. In order to capture the PSTs’ practices, regarding the use of MR to teach kinetic energy, the Preparatory Study (sessions 1 to 8, Figure 2), the Lesson Planning (sessions 9 to 11), the Research Lessons (sessions 12, 14, 16) and Reflections (sessions 13, 15, 17) were video recorded. In addition, the three educational researchers also participated in all sessions to take field notes. At the end of the LS, the PSTs were individually interviewed.

The Authors revised the English language as follow:

Table 2. Categories that emerged from data analysis

Category

Description

1. Changes on the MR that PSTs use to represent car crash-tests as a “real” world event.

2. Changes on the MR that PSTs use to support the teaching of the scientific concepts.

Evidences that PSTs changed the car crash-tests representations, namely PSTs started using authentic data and other realistic information, instead non-realistic data and frictionless crash-tests

Evidences that PSTs changed the kinetic energy representations, namely PSTs started using a sequence of photos of the dummy head in different instants of the collision, a table elaborated by the students and a Cartesian graph with experimental data, instead algebraic equations and other complex representations, such as Cartesian graphs with fictional data

Reviewer Comment: Line 194 and 199: Content analysis usually uses predefined categories while grounded theory seeks to define them through iterative comparison of the data. The authors should describe in a clearer manner how they combined these two approaches that they refer in lines 194 and 199 respectively.

Authors’ response:

Accordingly Reviewer’s remark, the Authors amended the text in the manuscript, on page 7 of the “3.3. Data Collection and Analysis” section, second paragraph, and changed the reference [48], as follows:

·         The video records of sessions 1 to 17 (about 51 hours in total) and the interviews (about five hours in total), were transcribed, and together with field notes of all sessions were analyzed to find changes that occurred in the PSTs’ practices regarding the use of MR. For this purpose, a descriptive and content analysis [47] was performed for qualitative data. In particular, the transcripts of video records and field notes were read by the first author of this article. After that, the targeted text was segmented, representing ideas related with the research aim. Each segment was assigned a code and clustered on a category, according to its features [48].

Reviewer Comment: Line 214: The difference between the two categories in Table 2 is not sufficiently clear neither in the method nor in the results section.

Authors’ response:

The Authors agree with the Reviewer´s remark. Thus some changes were made in order to clarify the difference between the two categories, on the Table 2, on page 8.

Category

Description

1. Changes on the MR that PSTs use to represent car crash-tests as a “real” world event.

2. Changes on the MR that PSTs use to support the teaching of the scientific concepts.

Evidences that PSTs changed the car crash-tests representations, namely PSTs started using authentic data and other realistic information, instead non-realistic data and frictionless crash-tests

Evidences that PSTs changed the kinetic energy representations, namely PSTs started using a sequence of photos of the dummy head in different instants of the collision, a table elaborated by the students and a Cartesian graph with experimental data, instead algebraic equations and other complex representations, such as Cartesian graphs with fictional data

Reviewer Comment: Lines 317 and 318: It should not be in the main text as they are a part of the task following figure 4.

Authors’ response:

The Authors agree that the lines referred by the Reviewer should be a part of the task. Thus the figure 4 is now as follows:

Considering the following list of algebraic equations, select the one from which you can determine the kinetic energy (E) of the car from its mass (m) and speed (V):

i)                          

ii)

iii)                    

iv)

Select the option that correctly related the kinetic energy (E) of the car with its mass (m) and speed (V).

Reviewer Comment: Line 476: In the discussion section it is mentioned that the PSTs were encouraged to deepen their knowledge on the use of MR. More information about the way they were encouraged by the research team is needed, in order to understand the procedure that took place helping PSTs to readjust their teaching practices using MR.

Authors’ response:

Accordingly the Reviewer’s suggestion, the Authors add more information in the text of the manuscript, on page 15, “Discussion” section, first paragraph, as follows:

·         The aim of this research was to investigate changes, promoted by the LS with three cycles, in the PSTs practices regarding the use of MR in teaching kinetic energy. During the LS sessions, the PSTs were encouraged to deepen their knowledge on the use of MR as well as to prepare, teach and improve their practice applying MR. The support for PSTs by the teachers educators and researchers was guided through an inquiry based strategies.

Reviewer Comment: 1.1) Lines 515-534: Explanations about the way the Lesson Studies were implemented, in comparison to other research using the same or similar methodology could be mentioned in the method section, since the way they are connected to the discussion of results is not clear.

1.2) The authors need to make a more clear connection between result and cause in the discussion, justifying why from Research Lesson 1 to Research Lesson 2 (i.e., end of the cycle 1 to cycle 2) occurred changes both in the MR of the event (Category 1) and in the MR of the scientific concepts (Category 2) while from cycle 2 to cycle 3 only occurred changes in the MR of the scientific concepts and possibly discuss it with other similar or contradicting research findings.

Authors’ response:

1.1). The Authors agree with Reviewer’s remark. Thus, more text was added in the “Method” section, particularly on “3.2 Lesson Study Implementation” section, sixth paragraph, as follows:

·         As already mentioned, this investigation included eight sessions for the Preparatory Study (sessions 1-8), and three sessions (sessions 9-11) to perform Lesson Planning (Figure 2). This option helps PSTs to gain knowledge and insight into science as well as about student thinking [38, 52]. The three cycles of the LS allow each one of the PSTs teach a Research Lesson. Although the repetition of cycles is controversial in LS [38], in initial teacher education as been showed beneficial [41]. This investigation intended to follow the original Japanese LS [53] as much as possible, as it had to be adjusted to the PSTs teacher education.

1.2) The Authors agree that the text mentioned by the Reviewer needs clearer. Thus, more text was added in the “Discussion” section of the manuscript, on pages 16-17, sixth paragraph, as follows:

·         Since there were changes over the three cycles, (i.e., from cycle 1 to 2 and from cycle 2 to 3) (Figure 2) one may infer that the three cycles of the LS had a positive effect on the results obtained in this investigation. However, from Research Lesson 1 to Research Lesson 2 (i.e., end of the cycle 1 to cycle 2) occurred changes both in the MR of the event (Category 1) and in the MR of the teaching of scientific concepts (Category 2) while from cycle 2 to cycle 3 only occurred changes in the MR of the teaching of scientific concepts. That occurred because from cycle 2 to cycle 3 the MR of the event (Category 1) remained successful.

Reviewer 3 Report

Thanks to the authors for giving me the opportunity to review their paper. The subject is interesting and useful in education.

The authors need to improve the paper, the general impression being that of negligence in writing the paper. In line 21 there is an inexplicable and incomprehensible figure. The bibliography is improperly formatted and is difficult to follow in the body of the article with references to the bibliography section.

The research carried out by the authors allows them to analyze the collected data and after their processing and interpretation, to conclude on the multiple representation in teaching. However, the authors do not take into account the fact that their study may have limitations. Is the study valid? Does the study have limitations? What could be the limitations? Can the 51 hours of video recording be considered representative?

Serious clarification is needed on this issue.

Author Response

AUTHORS’ RESPONSE TO REVIEWER

Lesson study as a mean to change secondary preservice physics teachers’ practice for the use of multiple representations in teaching

The authors would like to thank the time and constructive criticisms of the Reviewers of this manuscript that strongly contributed to its improvement.

The authors have addressed each of the Reviewers queries/comments as outlined in Table 1. In this table, the reviewer’s comments and suggestions are presented with a grey background whereas the authors’ responses have white background. All the modifications to the original manuscript are in track changes.

Table 1 – Answers to Reviewer #3

Reviewer Comment. 1.1) The authors need to improve the paper, the general impression being that of negligence in writing the paper. In line 21 there is an inexplicable and incomprehensible figure.

1.2) The bibliography is improperly formatted and is difficult to follow in the body of the article with references to the bibliography section.

Authors’ response:

1.1) The Authors thank you the possibility to clearer the revised manuscript.

The figure in between the “Abstract” and the “Introduction” section refers to the Graphical Abstract of the manuscript. The Graphical Abstract was designed to be a unique, compact, and visual summary of the all main aspects of the manuscript. The main intention is to capture the attention and interest of the reader to the manuscript content in an easy and fast way. Therefore, the Graphical Abstracts are typically placed by the editors at the top of the manuscript. In order to clearer the function of this figure, the Authors placed in the revised manuscript the title “Graphical Abstract” at the top of the figure.

1.2) The Authors agree with the Reviewer’s remark. The references were amended and now following the MDPI format.

Reviewer Comment. The research carried out by the authors allows them to analyze the collected data and after their processing and interpretation, to conclude on the multiple representation in teaching. However, the authors do not take into account the fact that their study may have limitations. Is the study valid? Does the study have limitations? What could be the limitations? Can the 51 hours of video recording be considered representative?

Authors’ response:

Authors agree with Reviewer’s comments. Thus add more information in the “Discussion” section, on page 17, last paragraph, namely:

·         The current investigation was performed using the LS with an adaptation (namely, the adjustment of the LS sessions’ content to the previous PSTs’ knowledge) and involving a small number of PSTs participants (n=3). In addition, the LS implementation used a specific topic (in concrete, the kinetic energy) that is taught to 9th grade students. Thus, the findings of this investigation are valid for the present results. The conclusions’ generalization to a wider range of scientific topics (physics and chemistry) and for all PSTs may be, partially or totally, invalid.

It remained for future research studies to provide a deeper understanding of what other elements of the LS with PSTs may contribute to enhance their learning regarding MR, and to other fields related to teachers’ professional knowledge. Studies with PSTs in LS using a larger number of groups are to be pursued in future investigations.

Round 2

Reviewer 2 Report

The manuscript substantially improved according to the comments.

There are still some points that could be improved.

Please see the attached file. The new comments are marked with yellow colour.
